# Education for Citizenship: The Meanings Chilean Teachers Convey in the Neoliberal Context

Silvia Redon Pantoja [1], Natalia Vallejos Silva [2] and José Félix Angulo Rasco [3],*

1 Lead Investigador Area Citizenship and Educaction, Centro de Investigación para la Educación Inclusiva SCIA ANID CIE160009, Pontificia Universidad Católica de Valparaíso, Valparaíso 2561427, Chile; silvia.redon@pucv.cl
2 Centro de Investigación en Educación (CIE), Universidad Bernardo O'Higgins, Santiago, Chile; natalia.vallejos@ubo.cl
3 Lead Investigator Area Evaluation and Currículum for Inclusivity, Centro de Investigación para laEducación Inclusiva SCIA ANID CIE160009, Pontificia Universidad Católica de Valparaíso, Valparaíso 2561427, Chile
* Correspondence: felix.angulo@pucv.cl

**Abstract:** This article presents the results from research into education for citizenship in which Chilean teachers participated. Ninety-nine interviews and two focal groups that included questions on knowledge, beliefs, values and practices related to education for citizenship were carried out. NVivo 12 software was used for the analysis of the discourses, following the direction similar to the grounded theory that considers elaborating free nodes, structuring categories and configuring categorical trees, according to the school's administrative dependency. The results yield six macro-categories: School, Authoritarianism, Neoliberalism, Curriculum, Teacher Role and Citizenship. The present article analyses the Neoliberalism macro-category formed, in turn, by the following subcategories: (1) subject and resistance, (2) competitiveness and individualism in a subject that is instrumental, consumer and reproducer of the establishment, (3) commodified schools, where the economic value regards students and families as clients, (4) a culture of bureaucratization and accountability, and (5) lack of a sense of communality as a collective, supportive body. In all of them, teachers show themselves eloquently critical of the neoliberal system and of the obstacles it poses to rights, justice and democracy in the current capitalist citizenship and school.

**Keywords:** education for citizenship; teaching staff; neoliberalism

## 1. Introduction

This article is framed within the theme of education and citizenship. The concept of citizenship puts into play fundamental values of society and therefore of education. On the one hand, the subject of law and on the other, the political community and the link between the two. Along with the relevance of its philosophical content, the historical, political and social context explains the emphasis on putting this concept at the center. At the end of the 80s this concept began to circulate in the academic community and from the years 1990 and 2000, there was an unprecedented explosion in the field of theoretical production in education and citizenship (not only is there a growing theoretical production, this emergence is also manifested in the publication of results from empirical research [1–36]) [37–59].

This theoretical explosion is explained, in turn, in relation to the social, political, economic and cultural changes that take place in the aforementioned decades, among which it is worth highlighting the following: the collapse of the welfare states [60]; the impact and challenges generated by transformations in the sphere of the globalized capitalist economy, together with information and communication technologies; the crisis and delegitimization of representative democracies as systems of government (in Western Europe and America) [61–63]; the processes of exclusion, inequality, poverty and inequity that affect countless people in the world; the emergence of "rejections, xenophobias and new

categories of citizens" [64] (p. 97) in contexts of growing multiculturalism, migration and loss of territoriality of the spaces of belonging; and, finally, the failure of environmental policies [59,65–67]. The triumph of the world capitalist system that has weakened the Welfare States under the neoliberal thrust requires rethinking education for citizenship in contexts of inequality and exclusion [62,68–72].

The current political, economic and social context is strongly determined by neoliberal logic at the global level, an issue that, in the case of Chile, was exacerbated in the years of dictatorship after the civil-military coup d'état (1973–1990) [73,74]. The thirty years of post-dictatorship governments (1990–2021) disguised the installed model with the face of democracy, reproducing logics of privatization of social rights and common goods [75,76] accentuating inequality in the population, at levels that the OECD ranks Chile as the second most unequal and segregated country in the world [77]. It is important to highlight that neoliberalism is much more than an economic model because it operates as a guiding rationality, it penetrates the subjective, social, institutional and state fabric, destroying the social bond, solidarity cooperation, collaboration and the deep sense for the common. [71,78–86]. Neoliberalism recognizes and protects individualism of the entrepreneurial, meritocratic, successful and competitive subject supported by instrumental cost-benefit logics aligned and focused exclusively on economic growth. In this framework, the market as a unique episteme subtly dispels the values that sustain the community bond, social rights, the dimension of the public and common sense [87–89].

The permanence, consolidation and impact of neoliberalism in the educational field in Chile, has been possible to a large extent by the structural process of privatization and by the installation of the New Public Management (NGP) governance [90–98], who have pointed out to match, in structural and functional terms, educational institutions with business culture, through the promotion of institutional (re)culturalization and redefinition of educational work, "which encourage subjects and institutions to model and guide in terms of standards, indicators and evaluation systems associated with public accountability" [99] (p. 34). This would explain why attempts to consolidate democratic and subject-centered pedagogies have had a very minor impact at the level of the Chilean educational system [100].

Education for citizenship, therefore, constitutes an important pedagogical challenge, since neoliberal educational policies do nothing but reduce and minimize the deep and complex meaning of education. There is an emptying of humanistic and academic content, as well as of the political, due to the loss of the sense of the common [87], stripping and canceling the livelihood and the core of society in its bond community [70,71]. Neoliberalism promotes and is expressed through educational policies that reduce the quality of education to management and accountability, in the primacy of quantification as a criterion of truth and the minimization of the educational process to learning outcomes of measurable behaviors in students [73,101–105] and in teacher performance evaluations [106]. Neoliberal educational policies not only contribute to weaken the reflective rationality of teachers and their autonomy and professional identity in practice [107,108], but also weaken their understanding as a political subject for social transformation and construction of democratic school cultures [109–111], and inevitably condition their knowledge about education for citizenship in school.

The research work that is presented in this article investigates the teaching conceptions about citizenship education in Chile, within the framework of the Chilean educational public policy that through law 20.911 (year 2016) (The Law 20.911 promulgated in 2016 establishes as mandatory for all educational establishments recognized by the State the development of a National Citizenship Education Plan that covers the levels of preschool, primary and secondary education, and that adjusts to the nine objectives that it establishes the law. Similarly, the regulation contemplates the incorporation of citizenship education in the initial training of teachers and the creation of a compulsory subject of Citizenship Education in the school curriculum in the last two levels of secondary education) that establishes the obligation to create, for all establishments officially recognized by the State,

a Citizen Education Plan (CEP) and teach the subject of citizenship education in the last two years of secondary education (from 2020). The Citizen Education Plan seeks that education for citizenship does not remain encapsulated in a subject, but rather (precisely) transcends the limits of a subject-matter and be embodied in practices and daily experiences that guide the actions of teachers and the various members of a school community. Therefore, analyzing the conceptions of citizenship in light of the mandate of public policy that is contained in the law of citizen education is relevant for understanding Chilean education. The historical trajectory of education for citizenship in Chile has privileged the teaching of civics as a structure and procedure of politics and has made the recognition of all educational events as political events invisible [112,113]. For this reason, collecting the teaching voices and their conceptions of citizenship education is enormously relevant to understand the normative and ethical-political ideology that underlies the teaching conceptions, an ideology that configures and is reflected in schools.

This article begins with a discussion about the concept of citizenship from its dynamic, open and necessarily situated dimension. Next, the field study becomes explicit with the epistemological-methodological section, the criteria for the selection of the participants and the analytical processes. The last section discusses the categorical results of the teachers' discourses and their meanings of the emerging subcategories of neoliberal-ism, organizing the discursive lines from a dual hermeneutics that links discourses to the political, social and pedagogical theory of education and the neoliberal context.

## 2. Education for Citizenship from a Situated Perspective

Etymologically, the concept of citizenship comes from the Greco-Latin *cives* and *polis*. The norms that regulate the coexistence in a common territory orchestrate two fundamental principles: on the one hand, the individual rights and, on the other, the notion of association with the community [59]. The theoretical density and diversity that presuppose the meanings of the concept of citizenship of the human being as having inalienable rights and of the political, juridical, and social community constitute its polysemous, equivocal and open dynamism in its conceptual approaches. From a situated perspective on the concept, we would like to make a distinction as to its epistemological dimension. The construction of the concept from an ontological, normative, closed and idealistically absolute rationality takes for granted a formal equality, abstracting itself and distancing itself from the real inequality [59]. Another epistemological perspective is hermeneutical-critical, and this helps us comprehend this concept situated in the world of life by assigning it meaning from history, culture, territory, and axes of power, in particular economic, which underlie the condition of citizen.

Usually, the theoretical discussions and approaches that develop the concept of citizenship do not refer to its social, economic, and political context. This omission is not innocent, since when separating the ideas about or the theoretical reflection on the temporal space, they get uncritically encapsulated, thus originating a kind of simplification, reduction and even a contradiction of the concept itself [114]. This epistemic practice—dissociated, fractured or separated from the temporal-space that contains it—requires enhancing and detailing precisely those concepts that provide said concepts with sense and meaning. The theoretical condition and perspective when naming, conceptualizing and constructing closed and stagnant 'ideal' realities are observed in the separation and breakup between signifier and signified. Foucault [115], in referring to language, poses the problem about the rupture between signifier and signified:

> The profound belonging to the language and the world has been destroyed. The prevalence of writing is over. This uniform layer where what is seen and read, the visible and the enunciated became indefinitely intertwined disappears. Things and words are going to become separated. [115] (p. 50).

The breakup between signifier and signified, between the words that enunciate, and the enunciated things confront, not in a trivial way, with the cultural importance of the functionality of objects according to their experience, just as Baudrillard points out [116].

Such importance, as it were, is granted by a context that establishes values, order and hierarchies of wants and needs [116]. This scenario, in which language already possesses a symbolic, interpretative and subjective load, provides a prelude, with all the more reason to make such a context explicit, a context that gives sense to the words as signifiers. Put it another way, it is not the things in themselves, but they 'are' according to the context where they find themselves. The signifiers, from this perspective, can take two paths: on the one hand, to reflect on the content of the vital reality that they attempt to represent, as guises of life and, consequently, of the space-time, culture and society coordinates that form them or, on the other, to distance themselves and become fractured, like timeless capsules, driven away from the life that shapes them. For this reason, to deal with education for citizenship, with politics or democracy, it is necessary to take into account the contexts (subjective, normative, socio-cultural, political, etc.) where those signifiers are signified.

Because of the above, when speaking of citizenship and education, the way in which the capitalist and neoliberal system has subjectivized coexistence, thinking, acting and feeling styles cannot be obviated. Citizenship, understood as that manner of democratic coexistence in a common territory, is regulated by the normative structure that determines in such coexistence what is 'right' and what is 'wrong' in juridical terms—what we usually call 'constitution'. And it is here, in the constitution, where we find the most serious uncritical fracture between 'text' and 'context'. Carlos de Cabo Martín [117], who makes an excellent analysis of critical constitutionalism, reminds us that most western constitutions, drawn on the English Magna Carta and afterwards the French one, in their basic structures, have not questioned the liberal and safeguard foundations of the private property with which they were created, with slight cosmetic modifications that are not concerned with questioning the XVIII-century society and individual versus the XXI-century capitalist neoliberal society. De Cabo Martín creates some analytical categories for explaining the obsolescence of the constitutions in effect today, such as the absolute normative and juridical codes in the coexistence of human beings of the XXI century. One of these analytical categories mentioned by Cabo Martín is the ethical-epistemological dimension, where all construction of scientific knowledge and its validation must consider an ethical telos of the wellbeing or of the well-living, associated with the emancipation of the collective liberation in the context of social justice. Another relevant category is historicity as a dimension that helps us revisit and deconstruct the present absolutized by the euro-centred colonialism that structures, organizes, regulates and univocally appraises the truth, the beauty and the good [117].

For this reason, the conceptualization of citizenship that regulates its coexistence in the juridical and normative structure of constitutions is not univocal; on the contrary, for some persons in power, 'becoming a citizen' was established as the standard bearer in the colonial fight and, for the indigenous peoples in Chile, it brought about the erasure, extermination and annulment of their identity culture. Just as in the past, citizenship presents itself as contradictory according to the individuals who call themselves citizens; today the tension is still latent between those who are part of it and those who live 'outside', those who are not part, nor do they have the right to have rights [118,119].

In this sense, citizenship is a dynamic and socio-historic category that is formed within the historic tension that the capitalist social order creates, caused by the struggle between the dominant sectors and the social movements that, curbed by the discipline and expropriation of vital processes—a feature of proletarianization—resist and come to be despite the narrow margin granted to human coexistence by the market [120] (p.14).

This is why citizenship is not a static concept; it cannot be a theoretically trapped and encapsulated signifier in an atemporal, ahistoric way and divested of its context. Normative conceptualisations of logics that refer to 'paper' memberships or euro-centrist juridical 'absolutes' operate in one way for some sectors of the citizens, but they simply do not exist for others. Those who insist on embracing and adhering to the conceptualisation of citizenship from rigid and normative assumptions undoubtedly simplify the uncritical

meaning separated from life, believing that knowledge-making is neutral and that an ethical-political dimension does not underlie the epistemology [121–123].

## 3. Method

The methodology used is based on a qualitative-hermeneutic epistemology focused on understanding the meanings in the professors' discourses around citizenship education. The understanding of meanings emerges ontologically and methodologically from an epistemology that does not seek to measure or quantify. For this reason, the study unit is based on a sample reasoned by criteria, which respond to the necessary heterogeneity versus homogeneity of the particularities investigated. These variables are expressed in the demographic characterization of the study unit belonging to 1238 educational establishments in the Valparaíso Region, Chile. Ninety-nine interviews and two focal groups (110 teachers) made up the field of study; the sample was selected according to gender, age, dependency on the school and discipline criteria, in conformity with how likely the teachers were willing to participate in the research. Interview protocols and focal group scripts were supported by a theoretical matrix set up around the following dimensions: (1) questions on identification or situated utterances: gender, age, type of school (administrative dependency), discipline they teach and administrative position at the school; (2) questions on the diagnosis of teacher training in education for citizenship; (3) questions on teachers' role and pedagogical tasks of education for citizenship; (4) questions on citizenship practices in the school (curricular contents, didactics and assessment); (5) questions on specific knowledge about citizenship, education for citizenship and public policies on citizenship; (6) questions on teachers' school life, their linking networks, acceptance of difference, government and democratic power handover.

The implementation of Law No. 20,911 (year 2016) in the Republic of Chile, establishes the obligation, in all educational establishments recognized by the State, to create a Citizen Education Plan. This means that teachers have to have the knowledge and didactic skills to create, develop and implement this education plan for citizenship in their educational communities, which is the basis for the problems of this research: What do teachers know, do and say to educate in citizenship? What meanings underlie citizenship education? What do they value in teaching citizenship?

### 3.1. Research Objectives

To respond to these questions, the following objectives are established:

- Know and understand the meanings that teachers build around education for citizenship.
- Identify the pedagogical practices that teachers value for citizenship education.
- Know the teachers' perceptions about the school culture and the experience of democratic coexistence in the school.

In order to respond to the objectives of this research and to ensure its internal structural coherence, the collection of information has been carried out through conversation (individual and collective) in the techniques of interviews and focus groups that aims to collect in depth the subjectivity of informants, from a hermeneutical epistemology.

### 3.2. Sociodemographic Characteristics of the Sample

Sampling reasoned by criteria: the 99 interviews and the 2 focus groups constituted the unit or field of the study (with 110 teachers in total), it was selected based on a sample reasoned by criteria of gender, age, dependence on the school and teacher discipline, according to the feasibility and willingness of people to participate in the research. Of the total of 110 teachers, 61 were female and 49 were male, which in percentages is made up of 55.4% women and 44.5% men. Of the total interviewed, the distribution by administrative dependency corresponds to 46.3% to schools with municipal-public dependency, 20.9% to private and private schools and 32.7% to subsidized or subsidized schools.

Characteristics of the sample according to educational level in which they work and gender: 36.3% of teachers work in primary education; 35.4% do so in secondary education

and 28.1% are teachers who teach both primary and secondary. When disaggregating the sample by gender, it is observed that 42.6% of women are teachers in primary education; 31.1% are teachers in secondary education and 26.2% develop teaching at both school levels. Regarding men, 28.5% are teachers in primary education; 40.8% in secondary education and 30.6% develop teaching at both school levels. Considering the dependence on the schools in which they work, 43.1% of primary education teachers work in municipal schools, and 22.2% of primary education teachers work in subsidized schools and 34.7% in paid private schools. Similarly, 39.2% of secondary education teachers work in municipal dependency schools; 38.8% do so in subsidized schools and 21.7% of secondary education teachers work in paid private schools. In the case of teachers who take classes in primary and secondary education at the same time, 17.6% work in municipal dependency schools, 38.8% work in subsidized schools and 43.4% work in paid private schools (see Tables 1–3).

**Table 1.** Distribution by grade and sex.

| Teachers | | | Women | | | Mens | | |
|---|---|---|---|---|---|---|---|---|
| Primary | Secondary | Primary & Secondary | Primary | Secundary | Primary & Secondary | Primary | Secondary | Primary & Secondary |
| 40 | 39 | 31 | 26 | 19 | 16 | 14 | 20 | 15 |
| | 110 | | | 61 | | | 49 | |

**Table 2.** Distribution by grade and administration.

| Public | | | Subsidized | | | Private | | |
|---|---|---|---|---|---|---|---|---|
| Primary | Secondary | Primary & Secondary | Primary | Secondary | Primary & secondary | Primary | Secondary | Primary & secondary |
| 22 | 20 | 9 | 8 | 14 | 14 | 8 | 5 | 10 |
| | 51 | | | 36 | | | 23 | |

**Table 3.** Distribution by age.

| Age | 25–34 | 35–44 | 45–54 | 54–64 | +65 | 25–34 | 35–44 | 45–54 | 54–64 | +65 |
|---|---|---|---|---|---|---|---|---|---|---|
| N° | 21 | 15 | 4 | 0 | 0 | 21 | 10 | 5 | 3 | 0 |

Distribution of the sample according to their age. The ages of the teaching staff are distributed as shown in Tables 4 and 5:

**Table 4.** Distribution by age and grade.

| Teachers | | | Women | | | Mens | | |
|---|---|---|---|---|---|---|---|---|
| Primary | Secondary | Primary & Secondary | Primary | Secondary | Primary & Secondary | Primary | Secondary | Primary & Secondary |
| 40 | 39 | 31 | 26 | 19 | 16 | 14 | 20 | 15 |
| | 110 | | | 61 | | | 49 | |

**Table 5.** Distribution by age and grade.

| Primary and Secondary Education | | | | |
|---|---|---|---|---|
| Age | 25–34 | 35–44 | 45–54 | 54–64 | +65 |
| N° | 14 | 16 | 1 | 0 | 0 |

50.9% of the teaching staff are between 25–34 years old; 37.2% are between 35–44 years old; 9% are between 45–54 years old; 2.7% are between 54–64 years old. Teachers aged 65 and over do not show up.

Distribution of the sample according to teaching specialty. The disciplinary specialties of the teaching staff are the following (Table 6):

**Table 6.** Distribution by disciplinary.

| Disciplinary Speciality | N° | % |
|---|---|---|
| History, Geography and Social Sciences | 28 | 30.8 |
| Natural Sciences (Biology, Physics or Chemistry) | 23 | 25.3 |
| Phlosopy | 7 | 7.7 |
| Arts | 5 | 5.5 |
| English | 10 | 11 |
| Language | 18 | 19.8 |
| Math | 12 | 13.2 |
| Religion | 3 | 3.3 |
| Physical Education | 4 | 4.4 |
| Total | 110 | 100 |

As can be seen in the Table 5, the majority of the sample corresponds to teachers who teach History, Geography and Social Sciences, Natural Sciences (Biology, Physics or Chemistry), Language and Mathematics.

*3.3. Techniques Used in Collecting Information*

The interview and the focus group are conversation techniques and not measurement instruments; therefore, they do not need to be validated as if they were questionnaires or another quantitative information collection system [124–129]. The interview protocol and the focus group script were supported by a discursive matrix organized around the following dimensions (1) situated speech linked to personal, academic and work biography (2) questions linked to formative processes in citizenship education; (3) questions related to the role of teachers in citizenship education; (4) questions about citizenship practices at school (curricular content, teaching and assessment); (5) questions related to specific knowledge about citizenship, citizenship education and public policy related to the latter; (6) questions oriented to the life of teachers at school, their networks, acceptance of difference, government and circulation of power within the framework of a democratic coexistence. The questions that are formulated from this thematic matrix do not operate as closed or structured questions, since the in-depth or ethnographic interview technique tends to follow the itinerary of the discourse through which the interviewee decides to travel. [130–136]. Even in the case of semi-structured interviews such as ours, the frameworks from which the questions can arise do not determine the process of the conversation or its evolution or development. The framework of the semi-structured interview does not operate as in a questionnaire, but rather indicates to the interviewer the basic themes from which they will ask her questions. But ultimately it is the responses of the interviewee, that is, the dynamics of the conversation itself, which clearly guides the entire interview process. As Brinkmann [137] emphasizes

> Compared to structured interviews, semistructured interviews can make better use of the knowledge-producing potentials of dialogues by allowing much more leeway for following up on whatever angles are deemed important by the interviewee; as well, the interviewer has a greater chance of becoming visible as a knowledge-producing participant in the process itself, rather than hiding behind a preset interview guide. And, compared to unstructured interviews, the interviewer has a greater say in focusing the conversation on issues that he or she deems important in relation to the research project. (p. 286).

The same occurs with the focus group technique: the itinerary of social discourses is constructed from the collective as a self-applied interview that emerges from the interviewees and not from the leadership of an outsider (the interviewer). [138–142]. As

Kamberelis & Dimnitriadis [141] (p. 325) stated "focus group allows researchers to explore group dynamics and the constitutive power of discourse in people's lives" and add:

> The leveling of power relations usually also allows researchers to explore group dynamics and the constitutive power of discourse in people's lives ( . . . ) The intensely social nature of focus groups tends to promote a kind of "memory synergy" among participants and bring forth the collective memory of particular social groups or formations [141] (p. 325).

*3.4. Qualitative Analytical Processes*

The analysis included three phases supported by the use of the NVivo 12 software (qualitative data analysis QSR International, Melbourne, Australia) and followed a path similar to the foundational theory [143]. The first phase was carried out by means of an inductive open codification that sought to synthesize and express the teacher discourses by classifying them and grouping them according to topics that shaped free nodes or first-order codes [144,145]. During the second phase, and once the free nodes were shaped, the structuring of the categories (axial and selective) was devised; these categories assume a more interpretative re-organization of the discursive text because their meaning is merely descriptive. This axial and selective phase was subject to a critical semantic analysis, binding the free nodes together on the basis of the thematic connections among them [146], and this gave rise to a categorical tree made up of six macro-categories: School, Authoritarianism, Neoliberalism, Curriculum, Teacher Role and Citizenship. Lastly, categorical trees were created according to the teachers' school administrative dependency (public state schools; subsidized, state-funded schools run by the private sector; and private schools). In total, 4368 free knots were found, which generates a great density of speeches, making it impossible to develop more than one category in an article. The choice of the Neoliberalism matrix is due to two reasons: on the one hand, the Chilean reality and the impact of this model on education for citizenship, and on the other, it is one of the most extensive and profound categories compared to the others.

## 4. Results and Discussion

Emerging from the teachers' voices, the Neoliberalism macro-category takes shape from various discursive lines linked to the following topics: (1) subject and resistance, (2) competitiveness and individualism in an instrumental consumer and reproducer of institutions, (3) commodified schools, where the economic value regards students and families as clients, (4) a culture of bureaucratization and accountability, and (5) lack of a sense of communality as a supportive collective body. These discursive groups—linked to the marketability of education, the bureaucratization of the entrepreneurship in the school public administration, the figure of the person sometimes resistant and other times self-marginalized, competitive and individualistic—find theoretical correspondence with the works of [70,89,92,93,147,148].

When it comes to defining neoliberalism, we must accept that this concept is not univocal [89]; that is, it is neither singular nor constant in its discursive formulations and material practices [70], yet it can be identified or characterized with the aid of strategies that break apart the wellbeing state and that destroy the sense of the social and its collective body, altering and changing the concept of freedom into free market, deregulation of capital flows and privatizations of common goods, enhancing inequalities, an increasingly continued, manifested and extended exclusion and injustice, entrenched in daily lives and, obviously, in the school [70,86,89,149].

*4.1. Subject and Resistance*

The neoliberal machinery is a complex and difficult-to-resist tangle. Teacher discourses show ambivalence between the 'functionary's' obedience to the system and its reproduction tied up by the economic power and the critical reflection of the students who wish to

distance themselves from it owing to their conscience that allows them to show themselves as if on the margin of the neoliberal machinery:

> T: Look, the most critical, the most intelligent feel they are on the margin of society, they feel they want to be on the margin, they feel prisoners in the society, they feel they are seized by the system (Teacher interview, Coverage 0.33%).

> T: They are being forced to be one way and to act in one way and to live in one way, and that they feel caught in this system and they look at you and say 'but ok, look if you', they say 'you have to come here from this hour till that hour and you are dependent on a salary and if you don't work, something is going to happen to you, and you are indebted and I don't want to live like you' the kids tell you all that and they are kids that many of them have good qualifications because they are capable ... no, they don't feel part of it. (Teacher interview, Coverage 0.71%).

Teachers acknowledge and value a generation of critical students who are not willing to reproduce the model, just as they observe it in their own teachers. It is a generation that has been able to stage a popular revolution and effect substantial post-dictatorship political changes in the Chilean governmental agendas.

*4.2. Competitiveness and Individualism in an Instrumental Consumer and Reproducer of Institutions*

Teachers hold neoliberalism responsible for the weakening of the social fabric as a collective body. It seems that what is most evident in the neoliberalist logic is the destruction of the community spirit. In their words, the search for profit, competitiveness, individualism, distrust, absence of solidarity and making or producing in the terms imposed by the market nurture relationships and the coexistence in the society in general and in the school:

> T: Neoliberalism! affects enormously because ultimately neoliberalism leads you to competence, and we are not going to thrive together, but me myself alone! And the ideal is to thrive by myself because I'd better go out into the market to compete! Then ... there we have it with the PSU and etc.(The PSU is the University Selection Test. It was applied in Chile until 2019 to select students who aspire to enter higher education. In 2020 it was replaced by the PTU (University Transition Test). Then, of course, neoliberalism. And that's why I think I focus a little on the sense of community and as if I struggled a little with that competitiveness ongoing in the current system. Because ultimately competitiveness forces you to be a lone individual and it's good for you that the other is not doing well because you prosper further (Teacher interview, Coverage 1.21%).

> T: ( ... ) the formal, traditional system is a competitive system that makes me vie for grades, it makes me vie for the teacher's attention. Then I don't think of the other, but of myself and that makes me turn into an egotistically socially being; instead, when we do not vie for the grades so that the whole class can compete for the same thing, so that the whole class can communicate o can express themselves, when one as a teacher seeks that everybody acquires knowledge (Teacher interview, Coverage 0.89%).

Teacher discourses emphasise an image of students not valuing the Other, except when the Other's existence allows for the achievement of individual interests. Teachers are aware of that and, in some cases, they struggle to change back this situation, but they also feel they are trapped as if they were reproducers of a system where their own subjectivity gets involved and their own pedagogical tasks echo:

> T: Today we don't understand ourselves as a collective group, we understand ourselves as individuals, then what the individual does is 'I scratch my own back', things are gotten by ... and sometimes with no moral restraints, true? Then, individuals centred on their own egos, on their own selves, but when we think as a collective group, I depend on the other ... on what I am going to obtain, it gets linked to what happens to me in relation to the other, then today we have to think as a collective group and today it's difficult, today we rather think as individuals and we teachers are not exempt from that, then suddenly

*I'm worried about my grades, about my subject, about my time, and not about me as part of a group where we are all being trained . . . then I should ask myself, let's see 'how can what's mine touch the other?' My colleague is looking into this, would that have to do anything with me? Am I concerned with it? Does it interest me?' because if I'm interested, students are likely to be interested, if the students see that I share with the other and we do things together and they see it, they perceive it and they have to do it, then they will understand that they we live in a community, but if each of them lives in their square metre, what I'm conveying as a message is 'things are like that, that's the way you live your daily life, the real life' one gets out and finds a super violent world, like 'me, me, me, here I am, get out of my way', then it's, ok, is that the kind of society we want or in which we are interested? (Teacher interview, Coverage 1.46%).*

Under the neoliberal rationality, market patterns are the only ones acceptable to regulate the relationships people establish among themselves. Teachers acknowledge that the school, instead of educating, trains subjects to adapt to a consumerist world and reproduce the established order. The ministerial discourse about emphasising critical reflection is a performance; that is, an empty signifier whose real and profound meaning finds no place to fit in and is a close ally of the obedience that demands that actions be complied and 'executed':

*T: If I become pessimistic, I think more than citizens, attempts are being made to train a consumer; erm . . . but the idea, as I've been telling you, from the ministry the discourse is very well articulated; that is, the idea is for individuals to be critical, reflective citizens, aware of the moment they are living, to demand their rights, but ultimately, what do we want? Submissive consumers out there (Teacher interview, Coverage 0.54%).*

*T: Because I think that our education, today's education, has all children numb; that is, truly, it does not allow children to think, to let them develop their . . . their critical thinking, for them to come up with new ideas (Teacher interview, Coverage 0.72%).*

*T: Because now we have to say that all of us are immersed in a fast world, of 'not thinking much' and do and only execute (Teacher interview, Coverage 0.18%).*

*4.3. Commodified Schools, where the Economic Value Considers Students and Families as Clients*

In Chile, today, education is not a State-guaranteed social right; it rather operates as consumption goods under the wing of a constitutional principle (the subsidiary state) in effect since the ratification of the 1980 Constitution, which has been interpreted as the freedom of those who want to offer educational services so that they can charge for them plus the family's freedom to hire and pay—in conformity with their incomes—for the education they want their children to receive [150]. In other words, education functions as consumer goods [149,151–156]. Among other things, this has caused those who cannot pay for an allegedly quality and private education to attend state or public schools (totally state-subsidized, where mostly the poor attend), and it has caused those that can pay to attempt to send their children to either subsidized schools (those schools that receive funding from the state and the families and that educate middle-class students) or private schools (those that receive no state funds and educate higher income students). The destruction and the loss of prestige of public education and of its teaching staffs has been a phenomenon occurring in tandem with the privatization and neoliberalisation of the education in Chile [157], as these teachers' discourse expresses:

*T: Erm . . . then there was the disrepute brought into the public education. All of it arose from where the public school had no conditions to compete with this type of logic. Nor did it have that culture! Not we, the public school was not born on the market; today teachers go to school to find students. Not we, we were not trained to go to schools and advertise a product and hunt down students (Teacher interview, Coverage 0.85%).*

*T: So, the very fact of your having, at some point, a government that says that parents, erm . . . they have to watch over their children's education is almost consumer goods.*

*Therefore, that's where the gap opens up ( . . . ), but it has to do with you considering education consumer goods, as something that you have to pay for, so you can receive it, as something that the state does not guarantee, then parents are held responsible, in the same schools where they have to do wonders and the state washes its hands (Teacher interview, Coverage 0.85%).*

In marketing logic, and education being consumer goods, private and susidised schools make profits and have to compete with each other to capture buyers (parents and caretakers) of their services and number of students (enrollment) so that they can stay in the system:

*T: Because the principal doesn't see that, I asked her the other day if she saw who left the school and who didn't and she said no, that she saw those who, did not see who they were etcetera, but no, so many left and for her they were a number, so many left and so many arrived (Teacher interview, Coverage 0.43%).*

### 4.4. Culture of Bureaucratisation and Accountability

In teacher discourses, bureaucratisation of civic education is understood and verified from various dimensions. Firstly, it is possible to highlight those views that, even though it is considered a very relevant topic around student training, admit it is not regarded as an area of particular interest (particularly by the schools' board of directors), beyond the obligation to comply with the Ministry of Education's requirements and, more precisely, with Law 20.911, which demands all schools that they should have a civic education plan. In other words, education for citizenship is just another task that should be performed 'as possibly as it can be performed', but it has no impact on the conceptions, values or practices of citizenship and democracy in the school:

*Q: And do you think that education for citizenship matters in your school?*

*T: I'd say it does not; it is not emphasized as it should.*

*Q: Beyond the documents?*

*T: Beyond the documents, yes, no, honestly citizenship training is very scarce. They feel content with the children singing the national anthem all Mondays; that's enough. That the history teacher focuses on the units relating to that is enough already. I think it should be much, much more important. That is, we are education citizens and that's where we are failing (Teacher interview, Coverage 0.95%).*

According to this reasoning, teachers also admit that they perform the role of an 'enacter' for the public policy on education for citizenship, but that their task lacks transcendence because it is not associated with a profound, collective, dialogic and democratic reflection on the the sense and scope of the education for citizenship in schools [158]:

*Q: Of course, and in the school this, the topic of citizenship training, is it incorporated into the IEP (Institutional Educational Project)? That is, does the institution's citizenship training exist as a school project? Or, is it taken from what the government says and here it is simply adapted?*

*T: It is an adaptation, an adaptation that surges just for the sake of complying. Basically, you've got to comply with certain events or facts that the ministry might inquire into, such as sessions and a schedule is set up and that schedule that you get at the beginning of the year tells you 'on that date you've got to do such thing' (Teacher interview, Coverage 1.78%).*

Teachers' voices mirror a critical view on the educational system that limits and hampers their role and de-professionalizes their work [159–162]. Neoliberal policies based on the New Public Management (See [163–165] for an analysis of the New Public Management.)—that highlight school efficacy and efficiency criteria associated with the compliance with achievement levels and standards, and the school accountability mainly based on standardized test scores—cause teachers to feel not only downhearted,

stressed and demotivated but a loss of community sense as well [166–169]. Accountability translated into the prioritization of the SIMCE and the PSU forces teaching staffs to reduce curricular times to address objectives and/or learning contents on education for citizenship. Under Chile's current system for Quality Assurance of Education, schools that receive state funding do their best to obtain good learning results on the SIMCE (SIMCE stands for Measurement of Educational Quality System, test-like assessments administered at various school levels that measure knowledge of, skills at and attitudes toward the school curriculum.), which is, ultimately, the decisive factor for the viability of an educational project [161].

> *T: I think the SIMCE has damaged our society but I also understand that it is a way of . . . of measuring, but this oftentimes has schools take to working solely on this type of teaching and tests which, generally, create more competitiveness, more math classes, language classes, thinking that they will succeed in life and leaving aside other things, for example what I just old you, commitment to my leaning and to myself, and from there, commitment to the society (Teacher Interview, Coverage 1.0%).*

*4.5. Increase in Inequality and Absence of Sense of Communality*

Chile is a country with a considerable socioeconomic segregation [170] and shows the greatest inequality among the OECD countries [77]. Inequality gaps in the country becloud rights and ways of social living, placing economic and meritocracy principles above them, inarguably shattering a citizenship project sustained in the context of social equality:

> *T: There's a kind of relation between those structures of which we are part and ourselves; I think that is like the relation between citizenship and state, I am part of that. Yet being part of that does not mean we are all equal within that part; I think the concept of citizenship sometimes tends to...to generate like an idea of equality that isn't real; as if it were obvious that the system through the state is considerably unequal. I think the citizen sometimes generates the idea that we are all equal as far as rights and duties go, but we live in a considerably unequal world, considerably classist, racist, male chauvinist; therefore, the idea that we are all equal is false (Teacher interview, Coverage 0.67%).*

## 5. Conclusions

Educational policies on education for citizenship, enacted by a law (N° 20.911) that requires all schools to develop a civic education plan, do not conform to the political and social entanglement of neoliberal laws at the core of the Chilean education. In this context, it is necessary to remember that Chile, on a world level, has been described as the "neoliberal experiment" that was first carried out during the 1973 dictatorship and has until today been perpetuated by the different post-dictatorship governments, evidently in different modes [89,95,150,171]. The Chilean society, its social rights forfeited, and its bonding texture fragmented, has come to exacerbating inequality and the instrumental individualism housed in subjectivities, manifesting itself in schools as a contradiction to what the official curriculum and Law N° 20.911 establish, and the daily life of educational communities. The notion of citizenship understood as the power of communality, on the one hand, and democracy as a lifestyle on the other, is deprived of content. The neoliberal logic assumes an instrumental reason whose focus is on the individual, on accountability, on homogenization through standardized measurement as symbols of educational quality and on the gradual loss of teacher autonomy.

The teaching staffs participating in this research are fully aware of the obstacles that the neoliberal context poses in order to exercise the rights, justice and democracy in the current capitalist citizenship and, consequently, the difficult task of teaching it. The speech attributes of the interviewed teachers, with respect to the demographic, work and academic variables do not present great differences in the discursive lines that pass through this category.

What called our attention was the trenchant critique of the neoliberal system since such a topic was not explicit in the interview protocols nor does it appear in the official

discourse on education for citizenship, either in curricular documents or in school textbooks [172]. The fact that it is a dense category of discourses shows that the teaching staffs' critical reading of the world allows them to open up a crack for its transformation. 'The very fact that people are able to admit that they are conditioned or influenced by the economic structures also makes them able to intervene in the determining reality' [173] (p. 66). This finding turns out to be enormously relevant for the scientific community in the field of study of education for citizenship, since, in most of the curricular texts of citizenship training, in educational policies, texts and related essays, the concept of Citizenship is timeless and uncritical of the political and socio-economic context that affects and determines the way in which freedom and equality are lived within the framework of social rights that ensure a good life [174–176]. For this reason, it is important to make visible the contribution of teachers who maintain a critical and situated view of citizenship in their vital dimension [177–180].

Although the teachers acknowledge being trapped in authoritarian school government logics, without professional autonomy to generate self-government processes with greater radicalism, they assume a great theoretical and educational void in citizenship issues, because it has not been contemplated in their teacher education, nor have had spaces to reflect on the political component in education [181].

The school is the only space for obligatory associativity that is left for the education for citizenship and the construction of a "we" [182] that helps reread the world with hegemonic categories counter to the capitalist neoliberal system.

**Author Contributions:** Conceptualization, S.R.P., N.V.S. and J.F.A.R.; formal analysis S.R.P. and N.V.S.; investigation, S.R.P. and N.V.S.; methodology, S.R.P. and N.V.S.; project administration, S.R.P.; supervision, J.F.A.R.; writing—original draft, S.R.P.; writing—review & editing, J.F.A.R. All authors have read and agreed to the published version of the manuscript.

**Funding:** Center for Research in Inclusive Education SCIA ANID CIE160009. Chile. Area Citizenship and Education.

**Institutional Review Board Statement:** The study was conducted according to the guidelines of the Declaration of Helsinki and approved by the Ethics Committee of Pontificia Universidad Católica de Valparaíso-Chile code: BIOEPUCV-H 427-2021.

**Informed Consent Statement:** Informed consent was obtained from all subjects involved in the study.

**Data Availability Statement:** The data will be accessible at the request of the authors.

**Conflicts of Interest:** The authors declare no conflict of interest.

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
