# Peer review of "Education for Citizenship: The Meanings Chilean Teachers Convey in the Neoliberal Context"

_sustainability, doi:10.3390/su132313390_

Round 1

Reviewer 1 Report

The research seeks to identify and analyze the knowledge, beliefs, values and practices related to citizenship education. The study is ascribed to qualitative approach studies, with inductive coding and categorization (grounded theory), based on 99 interviews and the realization of 2 focus groups with teachers (110 teachers). Of the emerging categories obtained, it focuses its analysis on the category 'neoliberalism' and its 5 sub-categories: 1. Subject and resistance; 2. Competitiveness and individualism; 3. Commoditized school; 4. Bureaucratic culture and accountability; 5. Sense of community. 

Although the results obtained could be perceived as predictable, they could potentially increase the body of knowledge existing so far in this field. In order to increase the scientific quality and impact of this study, the following actions on the manuscript are recommended:

1. The review of the scientific literature is particularly brief, especially when the object of study (education for democratic citizenship) has been the subject of a very abundant scientific production at the international level. 

2. The objectives of the research are not incorporated, briefly and explicitly, just before the "Method" section, although they are mentioned in the brief introduction and review of the literature. Likewise, a new wording of the 'Introduction' is recommended, which does not begin with "The results here in respond..."; it is not usual to begin an introduction in this way.

3. The type of sampling carried out is not specified; only the general selection criteria are indicated, but not defined. For example, what does it mean to select teachers according to age or sex? In what sense is this selection made?

4. The script of the interviews and focus groups, described between lines 142 and 150, does not provide evidence of qualitative validity, beyond its enunciation. These evidences can be provided by calculating Fleiss' Kappa or Kendall's W concordance coefficient.

5. Was there only one coder? If more coders were involved, please include, through any of the above indices, their degree of agreement, in order to ensure accuracy and absence of scientific bias.

6. The "Method" section is too brief. Please write and structure the section according to international scientific research standards: sociodemographic description of participants; instrument or data collection technique (including corresponding validation evidence); design and procedure; and data analysis.

7. A somewhat more updated review of the scientific literature and a contrast of its results with those recently obtained at the international level is recommended. 

Author Response

  1. The review of the scientific literature is particularly brief, especially when the object of study (education for democratic citizenship) has been the subject of a very abundant scientific production at the international level. 

The literature review has been expanded and improved considerably. This review can be found both in the Introduction and in section 2. Approximately from lines 24 to 204. See also references 1 to 123 that indicate the quality and depth of the review performed. We have also greatly improved the introduction.

2. The objectives of the research are not incorporated, briefly and explicitly, just before the "Method" section, although they are mentioned in the brief introduction and review of the literature. Likewise, a new wording of the 'Introduction' is recommended, which does not begin with "The results here in respond..."; it is not usual to begin an introduction in this way.

A new section has been incorporated with the stated objectives. The new section is called: 3. Research Objectives. From lines 205 to 222

  1. The type of sampling carried out is not specified; only the general selection criteria are indicated, but not defined. For example, what does it mean to select teachers according to age or sex? In what sense is this selection made?

In the Methodology section, in addition to indicating that “The method used stems from a qualitative-hermeneutic epistemology in order to understand the meanings that teachers assign to education for citizenship”, we have dedicated section 3.1. to show the Sociodemographic characteristics of the sample, also displayed in 5 tables. We have also clearly specified in section 3.2. The techniques used in collecting information.

  1. The script of the interviews and focus groups, described between lines 142 and 150, does not provide evidence of qualitative validity, beyond its enunciation. These evidences can be provided by calculating Fleiss' Kappa or Kendall's W concordance coefficient.

We would like to emphasize that this research emerges from an hermeneutical epistemology that employs qualitative methodological strategies (not instruments) of individual and collective conversation (semi-structured interview and focus group). Therefore, the validity processes are not the same as those used with questionnaires or other research instruments that do require the use of the aforementioned calculation processes. The validity of this qualitative research responds to the rigor of the selection of the sample reasoned by criteria that integrates heterogeneity versus homogeneity of the informants. A second element of validity is found in the density of the information collected and in its saturation both in the interviews and in the analytical categories in the two focus groups. Finally, a third element of validity is the analytic-hermeneutical triangulation through the coding process discussed by the coders (5 in total).

  1. Was there only one coder? If more coders were involved, please include, through any of the above indices, their degree of agreement, in order to ensure accuracy and absence of scientific bias.

As we have just mentioned in the previous point, 5 coders have been used and their analytical results have been triangulated in discussion and debate sessions until the final category matrix is drawn up and agreed upon.

  1. The "Method" section is too brief. Please write and structure the section according to international scientific research standards: sociodemographic description of participants; instrument or data collection technique (including corresponding validation evidence); design and procedure; and data analysis.

The methodological section has been considerably expanded by adding the following headings:

4.1. Sociodemographic characteristics of the sample

4.2. Techniques used in collecting information

4.3 Qualitative analysis process

  1. A somewhat more updated review of the scientific literature and a contrast of its results with those recently obtained at the international level is recommended. 

As mentioned in point 1the literature review has been expanded and improved considerably. See, for instance, references 1 to 123 that indicate the quality and depth of the review performed.

Reviewer 2 Report

Dear authors,

This text is devoted to one of the important topics for the modern education system and society. The presented research might have practical and scientific value.
However, I would encourage you to be more meticulous in the way of presenting the second part of your paper, as it addresses a lot the part of explaining the meaning of concepts, but it does not look like a literature review section. I suggest you to make a profound analysis of similar studies in order to show what are the main findings in the literature concerning education for citizenship. How is it presented in other parts of the world?

Moreover, when reading the results section, the reader must obtain an explanation concerning the connection existing between the analyzed subject and the studied variables. For instance, why did you choose neoliberalism? The reader who is not familiarized with economic doctrines or schools of economic thought might have questions concerning the relevance of such an area, unless you explain why it is relevant.    

In addition, you need to show how the findings are in the same line or in a different line with some previous research conducted in the same field.

Author Response

The literature review has been expanded and improved considerably. This review can be found both in the Introduction and in section 2. Approximately from lines 24 to 204. See also references 1 to 123 that indicate the quality and depth of the review performed. We have also greatly improved the introduction.

The use of the Neoliberalism as category is explained as follow:

In total, 4,368 free knots were found, which generates a great density of speeches, making it impossible to develop more than one category in an article. The choice of the Neoliberalism matrix is due to two reasons: on the one hand, the Chilean reality and the impact of this model on education for citizenship, and on the other, it is one of the most extensive and profound categories compared to the others.

The importance of the results are explained as follow in the conclusions:

What called our attention was the trenchant critique of the neoliberal system since such a topic was not explicit in the interview protocols nor does it appear in the official discourse on education for citizenship, either in curricular documents or in school text-books [172]. The fact that it is a dense category of discourses shows that the teaching staffs’ critical reading of the world allows them to open up a crack for its transformation. ‘The very fact that people are able to admit that they are conditioned or influenced by the economic structures also makes them able to intervene in the deter-mining reality’ [173] (p. 66). This finding turns out to be enormously relevant for the scientific community in the field of study of education for citizenship, since, in most of the curricular texts of citizenship education, in educational policies, texts and related essays, the concept of Citizenship is timeless and uncritical of the political and socio-economic context that affects and determines the way in which freedom and equality are lived within the framework of social rights that ensure a good life [174-176]. For this reason, it is important to make visible the contribution of teachers who maintain a critical and situated view of citizenship in their vital dimension [177-180].

Round 2

Reviewer 2 Report

Dear authors,

Your paper was significantly improved and I congratulate you for this aspect.

I kindly suggest you to introduce the research objectives in the section dedicated to the methodology, as a very short section written only for the objectives does not look too professional.

Best regards,

Author Response

Following your kind suggestions, we have included the research objectives in the section dedicated to methodology with sub-heading number 3.1. Modifying the total numbering of all the epigraphs.